# PEG-Mediated Protoplast Transformation of *Penicillium sclerotiorum* (scaumcx01): Metabolomic Shifts and Root Colonization Dynamics

**DOI:** 10.3390/jof11050386

**Published:** 2025-05-17

**Authors:** Israt Jahan, Qilin Yang, Zijun Guan, Yihan Wang, Ping Li, Yan Jian

**Affiliations:** Key Laboratory of Agro-Environment in the Tropics, Ministry of Agriculture and Rural Affairs, Guangdong Provincial Key Laboratory of Eco-Circular Agriculture, Guangdong Engineering Research Centre for Modern EcoAgriculture, College of Natural Resources and Environment, South China Agricultural University, Guangzhou 510642, China; isratanu35@gmail.com (I.J.);

**Keywords:** *Penicillium sclerotiorum*, protoplast transformation, *GFP* tagging, metabolomic analysis, plant–fungal interaction

## Abstract

Protoplast-based transformation is a vital tool for genetic studies in fungi, yet no protoplast method existed for *P. sclerotiorum*-scaumcx01 before this study. Here, we optimized protoplast isolation, regeneration, and transformation efficiency. The highest protoplast yield (6.72 × 10^6^ cells/mL) was obtained from liquid mycelium after 12 h of enzymatic digestion at 28 °C using Lysing Enzymes, Yatalase, cellulase, and pectinase. Among osmotic stabilizers, 1 M MgSO_4_ yielded the most viable protoplasts. Regeneration occurred via direct mycelial outgrowth and new protoplast formation, with a 1.02% regeneration rate. PEG-mediated transformation with a hygromycin resistance gene and *GFP* tagging resulted in stable *GFP* expression in fungal spores and mycelium over five generations. LC/MS-based metabolomic analysis revealed significant changes in glycerophospholipid metabolism, indicating lipid-related dynamics influenced by *GFP* tagging. Microscopy confirmed successful colonization of tomato roots by *GFP*-tagged scaumcx01, with *GFP* fluorescence observed in cortical tissues. Enzymatic (cellulase) seed pretreatment enhanced fungal colonization by modifying root surface properties, promoting plant–fungal interaction. This study establishes an efficient protoplast transformation system, reveals the metabolic impacts of genetic modifications, and demonstrates the potential of enzymatic seed treatment for enhancing plant–fungal interactions.

## 1. Introduction

*Penicillium* is a widely studied fungus that has significant applications in biotechnology as well as medical and food industries [1]. *Penicillium sclerotiorum* is an important species of *Penicillium* that has significant biotechnological and medical and food industries applications [2,3]. This fungi is known for the production of various bioactive compounds, such as antibiotics and enzymes, with several industrial applications [4]. Furthermore, molecular genetic techniques allow for targeted transfer of DNA within fungal genomes [5], providing tools for better definition of conserved metabolic pathways. Protoplasting is a commonly used way to prepare a cell to undergo genetic modification in fungi. It has been more than six decades since the first reports on protoplast isolation from yeasts and filamentous fungi [6,7,8].

Fungal protoplasts are an important and central tool for both physiological and genetic research on fungi [9]. Protoplasts are fungal cells from which the rigid cell wall has been enzymatically removed, leaving the cell membrane intact; this wall-less state facilitates the uptake of foreign DNA during transformation [9]. This technology has especially been a game-changer for genetic transformation of fungi, which is mainly centering on selecting and altering the target genes to uncover the gene functions more efficiently [10,11], whereas the initial stage relies on the optimal protoplast preparation. In filamentous fungi, protoplasts can be isolated from either mycelium, germinated conidia or intact conidia, although protoplastization is more difficult in the latter case [12]. Protoplast preparation involves removing the fungal cell wall, typically using enzymes [13]. As a result, enzymatic methods to protoplast have been favored in most laboratories [14]. Depending on the fungal species, their growth environments, and growth patterns, the chemical compositions of the cell walls varied [15]. The digestive enzymes needed for degradation, including chitinases, cellulases, pectinases, and proteases, are determined by the cell wall structure [16]. Different species of fungi have different conditions for protoplasts preparation [17]. Currently, there are few studies on the protoplast preparation and regeneration of *Penicillium* species, and these need to be further investigated.

Protoplasts are essential methods for transformation protocols that use polyethylene glycol (PEG) [18]. PEG (polyethylene glycol) facilitates the fusion of foreign DNA with the fungal cell membrane by destabilizing the membrane surface, promoting DNA uptake into the protoplasts [18]. Fungal transformation can be performed using other techniques, such as the biolistic [19] or *Agrobacterium*-mediated approaches [20], although these methods often require specialized equipment or conditions, unlike the simpler PEG-mediated protoplast transformation. A particularly simple and non-instrumental transformation method has been established using PEG, which remains the most popular method of genetic transformation in filamentous fungi [18]. Using *GFP* as a molecular tool provides additional advantages to explore host–pathogen interactions and visualize in situ processes of infection [21]. The use of *GFP* in fungi is a particularly versatile approach for delineating host–pathogen interactions and image infection. Though *GFP* has been widely employed as a marker in numerous organisms, the expression of a fluorescent marker has not been reported in filamentous *P. sclerotiorum*-scaumcx01.

Building on this approach, *GFP*-tagged fungal strains can be used to study colonization dynamics and enzymatic actions in plant hosts like tomato. Tomato plants are an excellent model organism to study fungal colonization and enzyme activity as they are of both economic and agricultural importance [22]. Enzymatic treatments can increase the efficiency of colonization of vascular plants by fungi, by modifying the structure of the cell walls of plants, allowing the fungi to anchor and exchange nutrients [23]. Some fungal strains, such as *Penicillium oxalicum* [24], *Penicillium citrinum* [25], and *Penicillium bilaii* [26], can boost root growth in plants by increasing nutrient uptake and modifying root architecture in response to colonization. This phenomenon has been attributed to synergistic interactions between the fungus and the host plant that result in enhanced root growth. The use of *GFP* fluorescence enables detailed monitoring of fungal growth and spatial spread, as well as the effects on the metabolism of tomato roots at the fungus–root interface [27]. The integration of these strategies determines the synergism of *P. sclerotiorum*-scaumcx01 enzymatic activity and host response, providing new insights for improving the interaction between fungi and plants to enhance agricultural productivity.

In this study, we developed a system for *P. sclerotiorum*-scaumcx01, a transformation method of the biological investigation of protoplasts, not previously reported for this species. Protoplast-mediated transformation and metabolomic analysis enable an integrated approach for the study of fungal adaptation and host interactions. *GFP*-tagged strain scaumcx01 is an excellent model for studying interaction between fungus and host cells in vivo. This protoplast-mediated transformation system provides a new tool for functional genomics studies of *P. sclerotiorum*-scaumcx01, which will contribute to a better understanding of its molecular mechanisms. Moreover, the tagging with *GFP* would allow the study of the distribution of the fungal colonization which paves the path to future studies.

## 2. Materials and Methods

### 2.1. Media Culture

Potato dextrose agar (PDA): 40.1 g PDA powder in 1 L double distilled water; potato dextrose broth (PDB): 24 g PDB powder in 1 L double distilled water; TB3 broth: 3 g Yeast extract, 3 g casamino acid, 200 g of sucrose in 1 L water, 15 g agar for TB3 agar.

Osmotic medium: A total of 295.8 g of MgSO_4_·7H_2_O, 0.46 g of Na_2_HPO_4_, and 1.06 g of NaH_2_PO_4_·2H_2_O were prepared. MgSO_4_·7H_2_O was dissolved in 1000 mL of distilled water. Separately, no more than 50 mL of distilled water was used to dissolve Na_2_HPO_4_ and NaH_2_PO_4_·2H_2_O. After complete dissolution, the phosphate solution was combined with the MgSO_4_·7H_2_O solution. The pH was then adjusted to 5.8 using 1 M Na_2_HPO_4_, and the final solution was stored at room temperature for use.

### 2.2. Fungal Strain and Plasmid Vector

The endophytic fungus *P. sclerotiorum*-scaumcx01 from *Portulea oleracea* (GDMCCNo.60249) was used. The strain scaumcx01 was screened, identified, and preserved in the Guangdong institute of microbiology by the research group of the Chemical Ecology Laboratory, South China Agricultural University. For future use, certain strains were mixed with 50% glycerol in the ratio of 1:1, and stored in the refrigerator at −80 °C. The transformation vector pCT74-*GFP* used in this study was obtained from Dr. Huawei Zheng of the Plant Protection College, Fujian Agriculture and Forestry University. The pCT74-*GFP* vector is a 5.7 kb plasmid that contains the hygromycin resistance gene [28].

### 2.3. Protoplast Generation

#### 2.3.1. Hyphal Preparation

To prepare hyphae for protoplast isolation, *P. sclerotiorum*-scaumcx01 was inoculated on a PDA plate and incubated in the dark at 28 °C for approximately 5–7 days. Once spores are produced, 3–5 mL of a 0.1% Tween-80 solution were added to the plate. The spores were gently scrapped and transferred to a 50 mL centrifuge tube. The tube was vortexed for 5 min to ensure the spores were fully dispersed. Subsequently, the spores were washed three times using an osmotic stabilizer (osmotic media). After the final wash, the spores were filtered and resuspended in 100 mL of PDB and incubate at 28 °C and 180 rpm overnight to initiate germination. Hyphal development was monitored by taking samples at 16, 24, 30, 48, and 72 h. When the hyphae reached a suitable stage of development, they were harvested for protoplast preparation.

#### 2.3.2. Enzymatic Digestion for Protoplast

Germinated spores were harvested by centrifugation (4 °C, 4000 rpm, 10 min), washed three times with 15 mL STC buffer (1.2 M sorbitol, 50 mM Tris-HCl pH 8.0, 50 mM CaCl_2_, pH 7.5), and suspended in 20 mL enzymatic digestion solution. To optimize protoplast yield, the enzyme mixture (30 mg Lysing Enzymes, 20 mg Yatalase, 10 mg cellulase, and 10 mg pectinase) was dissolved sequentially in 20 mL osmotic medium, vortexed, and filtered (0.22 μm). Spores were incubated in this solution at 28 °C, 80 rpm for 12 h, then filtered through sterile 40 μm Miracloth, rinsed twice with STC buffer, and centrifuged (4 °C, 5000 rpm, 10 min). The protoplasts were resuspended in 6 mL ice-cold STC buffer, stored at 4 °C, and confirmed by microscopic examination. Vendor information and catalog numbers for the protoplasting enzymes used in this study are provided in Appendix A.

#### 2.3.3. Protoplast Yield Optimization

To determine the optimal conditions for maximum protoplast yield, the enzymatic digestion process was conducted under varying incubation times (e.g., 1 h, 2 h, 3 h, 6 h, 8 h, 12 h, 14 h, 16 h, and 20 h) and temperatures (e.g., 25 °C, 28 °C, 30 °C, and 32 °C). For each condition, 20 mL of the optimized enzyme digestion solution, prepared as described in Section 2.3.2, was added to the germinated spore suspension. The suspensions were incubated in sterilized 50 mL conical flasks placed in a constant temperature shaking incubator set to the respective conditions. Microscopic examination and cell counting using a hemocytometer were conducted to assess protoplast yield under each tested condition.

#### 2.3.4. Osmotic Stabilizer Test

To evaluate the effect of different osmotic stabilizers on protoplast yield, fungal mycelia were prepared by incubating spores in PDB at 28 °C for 24 h (see Section 2.3.1). The stabilizers tested included NaCl, MgSO_4_, KCl, Sucrose, CaCl_2_, and Sorbitol at equimolar concentrations (1 M). After enzyme treatment, protoplasts were washed with STC solution to remove residual enzymes and stabilizers. The protoplasts were then collected by centrifugation (e.g., 4000 rpm, 10 min), and the yield was determined using a hemocytometer. The stabilizer yielding the highest concentration of viable protoplasts was identified.

#### 2.3.5. Protoplast Regeneration and Viability Assessment

For protoplast regeneration, 200 µL of the protoplast suspension (2 × 10^6^ protoplasts/mL) was mixed with 10 mL of TB3 liquid medium in a 50 mL centrifuge tube and incubated on a rotary shaker at 28 °C and 80 rpm for 16 h. After incubation, molten TB3 solid medium supplemented with 50 µg/mL ampicillin was added to the cultures to prevent bacterial contamination, bringing the total volume to 50 mL. The number of regenerated and un-regenerated protoplasts was determined using a cell counter, and the regeneration rate was calculated based on the method described by Li et al. [29]. The plates were incubated at 28 °C for 7 days to allow mycelial colony formation. All experiments were performed in triplicate and repeated three times.

#### 2.3.6. Screening of Strain scaumcx01 Resistance to Hygromycin Resistance Gene

To determine the optimal hygromycin B (HyB) concentration for selection, the antibiotic was added to sterilized potato dextrose agar (PDA) medium cooled to approximately 45 °C, resulting in final concentrations of 0, 10, 30, 50, 70, 90, and 100 mg/L. After the medium solidified, 0.5 cm mycelial plugs of *P. sclerotiorum*-scaumcx01 were placed at the center of each plate. The plates were sealed with parafilm and incubated at 28 °C for 7 days. Fungal growth was monitored, and the hygromycin concentration that effectively inhibited colony development without completely suppressing growth was selected as the optimal screening concentration for subsequent transformation experiments

### 2.4. Genetic Transformation of Strain scaumcx01 Protoplasts

To initiate the transformation, 20 μg of the expression vector pCT-74-*GFP*, which contains both Amp and HyB resistance genes, was gently mixed with the protoplasts and allowed to stand on ice for 20 min. Next, 1.25 mL of PEG 8000 solution (20% *w*/*v* PEG 8000 mixed with STC buffer) was added to the mixture. After another 20 min on ice, the liquid was transferred to 20 mL of sterilized TB3 medium containing 50 mg/L Amp antibiotics, and the culture was incubated overnight in a constant temperature shaker at 28 °C and 120 rpm. The protoplasts were then plated on PDA medium, with the bottom layer containing 50 mg/L Amp and 50 mg/L HyB. After 2 h, the upper layer, containing 50 mg/L Amp and 70 mg/L HyB, was added. The plates were incubated in a constant temperature incubator at 28 °C for 7 days. After 2 days, the plates were flipped upside down to encourage proper growth. Following 7 days of incubation, fungal mycelium was picked and transferred to fresh PDA medium containing 50 mg/L Amp and 70 mg/L HyB. The transformants were screened twice on PDA medium containing these antibiotics, and those that successfully regenerated were considered positive transformants.

#### 2.4.1. Fluorescence Intensity Detection of Strain scaumcx01 Transformants

Microscopic observations were carried out using a Leica DMRBE microscope (Wetzlar, Germany). All objects were placed on a glass slide in a water droplet, covered with a cover slip, and observed without further manipulation. Light microscopy was performed using white light without any filters. Fluorescence Zeiss LSM800 Confocal Laser Scanning microscopy (Jena, Germany) was performed. LSM800 absorbs the appropriate amount of sample drops on the slide, followed by green light excitation (excitation wavelength 484, maximum emission wavelength 501 nm). Then, the instrument is preheated, booted, and the appropriate area to shoot is selected. Light microscopy was performed with the same microscope, without filters.

#### 2.4.2. Assessment of the Transformants

To verify plasmid integration, putative *GFP* transformants were assayed for transgene stability by transferring a Mycelium plug from the leading edge of a 1-week-old hygromycin-amended (70 mg/L HyB) PDA culture onto the edge of PDA without hygromycin detectable in 50 mm or 90 mm sterile Petri plates. The following four passages (second to fifth) were carried out on PDA without hygromycin to test the stability of the *GFP* marker under non-selective conditions. The sixth passage was then conducted on PDA medium containing hygromycin B (70 mg/L) to confirm stable integration and expression of the *GFP* gene. Fluorescence microscopy was used to assess the intensity and localization of *GFP* expression in fungal structures.

#### 2.4.3. Confirmation of *GFP* Presence and Localization

The stable *GFP* transformants were grown on PDA plates at 28 °C, and fungal mycelium was harvested for DNA extraction. Genomic DNA was extracted from both transformed and untransformed (control) endophytic fungi using a standard method with minor modifications, as described by Moller et al. [30]. To confirm the presence of the *GFP* and *hygR* genes, as well as to verify fungal identity, PCR was performed using gene-specific primers listed in Table 1. The *GFP* gene was amplified as a 720-bp fragment, the hygromycin resistance gene (*hygR*) as a 1020-bp fragment, and the fungal ITS region as a 140-bp fragment. PCR conditions included 35 cycles of denaturation at 94 °C for 1 min, annealing at 60 °C for 35 s, and extension at 72 °C for 30 s, followed by a final extension at 72 °C for 5 min.

### 2.5. Metabolomic Analysis of Wild-Type and GFP-Tagged scaumcx01

#### 2.5.1. UPLC-HRMS Detection for Preparation of Test Solution

Wild-type and *GFP*-tagged strain scaumcx01 were cultured on PDA at 28 °C for 7 days. Mycelia were scraped, flash-frozen in liquid nitrogen, and ground to a fine powder. One gram of mycelium was extracted in 10 mL ethyl acetate by vortexing for 10 min at room temperature. After centrifugation (10,000× *g*, 4 °C, 10 min), the supernatant was collected, the process was repeated twice, and the combined extract was evaporated at 40 °C under reduced pressure. The dried extract was reconstituted in 1 mL methanol, filtered (0.22 µm), and stored at −20 °C. For UPLC-HRMS, 100 μL of the sample was mixed with 400 μL ice-cold methanol, sonicated (ice bath, 15 min), precipitated (−20 °C, 30 min), and centrifuged (20,000× *g*, 15 min). The supernatant was transferred to an injection vial for analysis.

#### 2.5.2. Liquid Phase Mass Spectrometry Conditions

UPLC was performed using a Thermo Vanquish Flex UPLC system with an ACQUITY UPLC T3 column (100 mm × 2.1 mm, 1.8 µm, Waters, Milford, MA, USA). The mobile phase consisted of (A) 5 mmol/L ammonium acetate + 5 mmol/L acetic acid in water and (B) acetonitrile. The gradient elution program was: 0–0.8 min: 2% B; 0.8–2.8 min: 2–70% B; 2.8–5.3 min: 70–90% B; 5.3–5.9 min: 90–100% B; 5.9–7.5 min: 100% B; 7.5–7.6 min: 100–2% B; 7.6–10.0 min: 2% B; Flow rate: 0.3 mL/min; Column temperature: 40 °C. For HRMS, a Q-Exactive Plus mass spectrometer (Thermo Fisher, Waltham, MA, USA) was used in positive and negative ion modes. Ion source parameters: sheath gas 35, auxiliary gas 10, spray voltage +3500 V (positive) and −3000 V (negative), and source temperature 350 °C. Data acquisition followed the DDA mode with a primary scan range of 70–1050 *m*/*z*, resolution 70 K (@ *m*/*z* 200), AGC target 3e6, and max IT 100 ms. The top five most intense ions (>100,000 intensity) underwent MS/MS fragmentation (resolution 17.5 K, max IT 50 ms, dynamic exclusion 6 s).

### 2.6. Enzymatic Pre-Treatment of Tomato Roots for Fungal Colonization Analysis

Tomato seeds were surface sterilized for 1 min in 70% ethanol followed by 5 min in 2% sodium hypochlorite. Seeds were rinsed thoroughly with sterile distilled water, dried under aseptic conditions, and then germinated on moist filter paper in Petri dishes at 28 °C for 2 day. For enzymatic treatment, a 1% (*w*/*v*) cellulase (Sigma C1184, St. Louis, MO, USA) solution was prepared in sterile distilled water. Seeds germinated and having young roots were then incubated in the enzyme solution for 20 min with shaking (80 rpm, room temperature), followed by thorough rinsing with sterile water. *GFP*-labeled strain scaumcx01 spores were cultured on PDA at 28 °C and the harvested spores were suspended in a sterile 0.01% Tween 20 solution and adjusted to 1 × 10^6^ spores/mL using a hemocytometer.

For enzymatic pre-treatment, excised root segments were immersed in 0.5% (*w*/*v*) cellulase in sterile phosphate-buffered saline (PBS). Enzyme solutions were filter-sterilized (0.22 µm syringe filter) before use. Roots were incubated at 28 °C with gentle shaking (80 rpm) for 5–10 min, then rinsed thoroughly with sterile distilled water. Pre-treated roots were inoculated with the *GFP*-labeled fungal spore suspension (1 × 10^6^ spores/mL) and incubated at 28 °C under high humidity for 12 h. Fungal colonization and spore penetration were observed using confocal microscopy.

#### Confocal Microscopy Imaging for Root

Fungal colonization on the pre-treated tomato roots was observed using confocal microscopy. The *GFP* fluorescence (FITC) was excited using a 488 nm laser, with emission detected between 520–530 nm, while the Rhodamine (Rho) fluorescence was excited at 561 nm, with emission observed at 529 nm. A 10× objective lens was used for imaging, and the system was set to a high gain of 750 V to optimize fluorescence signal detection. Images of *GFP*-labeled strain scaumcx01 spore penetration and fungal growth were captured, and the fluorescence signals were analyzed to assess fungal colonization.

### 2.7. Statistical Analyse

Statistical analysis was performed using appropriate software (e.g., SPSS (https://www.ibm.com/spss), GraphPad Prism 9.4 (https://www.graphpad.com/) to evaluate the significance of the results obtained from various experiments. Data were expressed as means ± standard deviation (SD). One-way analysis of variance (ANOVA) was employed to assess differences between groups, followed by Tukey’s HSD to determine which groups were significantly different. A *p*-value of less than 0.05 was considered statistically significant. All experiments were performed in triplicate and repeated at least three times to ensure consistency and reproducibility.

## 3. Results

### 3.1. P. sclerotiorum-scaumcx01 Protoplast Isolation

The protoplast preparation was assessed in relation to the age of fungus as well as mycelium status, and temperature. The highest protoplast yield was achieved from liquid mycelium after 24 h of incubation at 28 °C, with a concentration of 6.61 × 10^6^ cells/mL. Protoplast numbers significantly decreased after 30 and 48 h of incubation (Figure 1a). This reduction in yield may reflect a thickening of the mycelial cell wall over time, which could hinder enzymatic hydrolysis. Furthermore, protoplast production was markedly lower at 30 °C and 48 °C, suggesting that elevated temperatures negatively impact protoplast generation (Figure 1d).

### 3.2. Effect of Enzyme Combination on Protoplast Preparation

Fungal sensitivity to cell wall lyases varies due to differences in cell wall composition. In this study, a combination of 30 mg Lysing Enzymes, 20 mg Yatalase, 10 mg cellulase, and 10 mg pectinase was used. As shown in Figure 1e, protoplast yield varied significantly across different enzyme combinations. The highest protoplast yield, 6.72 × 10^6^ cells·mL^−1^ at 12 h, was achieved with this enzyme mix, significantly higher than other treatments. A positive correlation was observed between protoplast yield and enzyme concentration, indicating that *P. sclerotiorum* exhibits selectivity toward different enzymes based on its cell wall composition. Enzymes likely target different sites in the cell wall, leading to variations in protoplast hydrolysis. Additionally, the highest yield occurred at 12 h, with a sharp decrease at 14 h, highlighting the importance of precise timing in enzymatic digestion (Figure 1c).

### 3.3. Osmotic Stabilizer and Other Factors on Protoplast Preparation

In addition to fungal age and the state of mycelium, several other factors such as osmotic stabilizer, enzymatic hydrolysis period, and temperature will also have a significant influence on the yield of protoplasts. In Figure 1b, the protoplast number of *P. sclerotiorum*-scaumcx01 in inorganic salt (NaCl, KCl, MgSO_4_, CaCl_2_) osmotic stabilizers was significantly greater than that in organic solution (sucrose and sorbitol) osmotic stabilizers. Very few protoplasts appeared under the organic solution as osmotic stabilizers, indicating that protoplasts had a more preferable maintenance of cell morphology in the situation that contain inorganic salts. Among the inorganic salt osmotic stabilizers, 1M MgSO_4_ resulted in the highest protoplast yield, surpassing NaCl, KCl, and CaCl_2_ at the same concentration. The highest protoplast yield of 6.03 × 10^6^ cells·mL^−1^ was obtained with 1 M MgSO_4_. After enzymatic digestion, the protoplasts were washed with STC solution to remove residual enzymes and stabilizers, ensuring clean and viable protoplast preparations for subsequent experiments.

### 3.4. Protoplast Regeneration

Protoplast release and regeneration were successfully achieved from *P. sclerotiorum*-scaumcx01 on the TB3 medium. While initial protoplasts were detected as early as 6 h post-treatment in preliminary experiments (Figure 1c), the yield was low and inconsistent at that stage. Therefore, 12 h was selected as the starting point for further analysis under optimized conditions. Protoplasts were consistently released from the hyphae at 12 h (Figure 2a), with increased release by 14 h (Figure 2b). Purification of the protoplasts was completed, as shown in Figure 2c. Upon release and purification, protoplast budding was observed, with synchronous budding occurring at specific intervals (Figure 2d). New buds continuously emerged from the protoplasts (Figure 2e), and regeneration occurred through two distinct mechanisms: protrusion from the side of the protoplast to form a new protoplast (Figure 2f) and direct growth of mycelium from the protoplast (Figure 2g). After 3–4 days, visible colonies formed as a result of successful regeneration (Figure 2h). The regeneration rate was calculated to be 1.02%.

### 3.5. PEG-Mediated GFP Transformation and Selection of P. sclerotiorum-scaumcx01

To evaluate the transformation efficiency of *P. sclerotiorum*-scaumcx01 protoplasts, we performed polyethylene glycol (PEG)-mediated transformation using the plasmid pCT74-*GFP*, which contains both a green fluorescent protein (*GFP*) reporter gene and a hygromycin B resistance (*hygR*) gene. The wild-type (WT) strain was first screened on TB3 medium supplemented with varying concentrations of HyB. It was unable to grow at 70, 90, or 100 mg/mL, while moderate growth was observed at 10, 30, and 50 mg/mL (Figure 3), confirming that HyB could be used as a selection marker. Following transformation with the pCT74-*GFP* plasmid, colonies appeared on PDA medium supplemented with HyB, indicating successful selection (Figure 4a,b). In contrast, the WT strain did not grow (Figure 4c). PCR analysis confirmed the presence of the hygromycin resistance gene in the transformants (Figure 4d).

To further verify successful transformation, all putative transformants were analyzed by PCR targeting the *GFP* gene, which yielded the expected 0.72 kb band in the plasmid control and transformants but not in the WT (Figure 4d). Moreover, *GFP* fluorescence was clearly observed under laser confocal microscopy (Figure 5), confirming expression of the transgene. It is worth noting that the *GFP* gene is located downstream of a NOS terminator and is not driven by a fungal-specific promoter (e.g., gpdA). As a result, *GFP* expression may be relatively low and context-dependent, which should be considered when interpreting the fluorescence results. The transformants retained genetic stability after subculturing on PDA without selective pressure.

#### Stability and Fluorescence Localization of GFP-Tagged scaumcx01

To assess the genetic stability of the *GFP* gene in *P. sclerotiorum*-scaumcx01 transformants, the first passage was grown on PDA medium containing hygromycin to ensure initial selection. The following three passages (second to fourth) were conducted on PDA without hygromycin to test stability under non-selective conditions. The fifth passage was then performed on PDA containing hygromycin. Stable transformants showed normal growth in the presence of hygromycin, confirming stable integration of the *GFP* gene into the fungal genome. The intensity and localization of *GFP* expression were assessed in fungal structures using fluorescence microscopy (Figure 5). *GFP* fluorescence localizes primarily in fungal spores, with lower fluorescence in mycelium. It is important to mention that highest fluorescence intensity was found in spores, suggesting strong expression of *GFP* in these structures. In hyphal cells, *GFP* expression was more prominent in young hyphae, while aging hyphae exhibited weaker fluorescence. Additionally, the fluorescence pattern suggested possible accumulation in vacuoles or other organelles, although precise subcellular localization requires further investigation (Figure 6).

### 3.6. Metabolomic Analysis of GFP Tagged Strain scaumcx01

Metabolomic profiling showed clear differences between the wild-type and *GFP*-tagged strain scaumcx01. Total ion chromatograms (TICs) in both positive and negative ion modes revealed visible changes in metabolite composition after *GFP* integration (Appendix A). Heatmap analysis highlighted several differentially abundant compounds between the two strains (Figure 7c). According to volcano plot analysis, 2502 metabolites were upregulated, 1570 were downregulated, and 913 remained unchanged in the *GFP*-tagged strain compared to the wild-type (Figure 7b). Principal component analysis (PCA) of three biological replicates showed clear group separation, confirming that *GFP* tagging affected the strain’s metabolome (Appendix A).

While these changes may be attributed to *GFP* integration, it is important to consider that other factors, such as the expression of the *hygR* gene and potential off-target mutations, could have contributed to the observed metabolic shifts. Pathway enrichment analysis revealed significant changes in glycerophospholipid metabolism, which was the most affected pathway based on KEGG annotation (Figure 7a). Glycerophospholipids are major structural components of fungal membranes and play essential roles in membrane fluidity, vesicle formation, and signal transduction. The observed shifts in this pathway suggest that *GFP* integration may disrupt membrane homeostasis or cellular signaling, thereby influencing overall lipid metabolism and broader physiological processes in *P. sclerotiorum*-scaumcx01.

### 3.7. Colonization of Tomato Roots by GFP-Labeled Strain scaumcx01

Building on the observed promotion of root growth by *P. sclerotiorum*-scaumcx01, further microscopic analysis revealed that the *GFP*-labeled strain successfully colonized the tomato roots of the local heirloom variety *Solanum lycopersicum* (Fenxing Old Tomato) following enzymatic treatment of the seeds (Figure 8). Although the fungus does not penetrate the root tissue, it likely interacts with the root surface, potentially aiding plant growth or facilitating nutrient exchange. The enzymatic treatment involved the use of cellulase, which likely played a crucial role in partially degrading the root surface barriers, enhancing fungal attachment and interaction. After the treatment, the *GFP*-tagged strain scaumcx01 was incubated for 12 h, and distinct *GFP* fluorescence was observed on the root surface and within the root tissues, indicating successful fungal attachment and colonization. The fluorescence was most prominent in the cortical regions of the roots, suggesting active fungal interaction and potential nutrient exchange in these areas. Fungal spores were applied at a concentration of 10^6^ spores/mL, and the enzyme pretreatment appeared to facilitate fungal access to the roots by altering root exudates or modifying the root surface, thereby promoting fungal recognition and signaling. These results demonstrate that enzymatic seed treatment, combined with an optimal spore concentration, is a viable method to enhance the interaction between tomato plants and strain scaumcx01, potentially supporting plant growth through surface-level colonization and nutrient exchange.

## 4. Discussion

Both in classical and molecular genetics, protoplasts are an important biological tool [31]. This highlights the need to develop an efficient and reproducible protocol for protoplast formation in the fungus under study [32]. Despite the fact that *P. sclerotiorum*-scaumcx01 is a recently discovered fungus with a desirable biocontrol mechanism, no protoplast systems have been created for the species. In the current study, we investigated key factors affecting protoplast yield and developed the conditions for protoplast preparation in strain scaumcx01. In line with earlier studies, fungal age and mycelium status were vital for successful protoplast generation [17,33,34]. Our findings showed that incubation at 28 °C for 24 h resulted in the maximum number of viable protoplasts (6.61 × 10^6^ cells/mL) (Figure 1a). Extended culture times were associated with a decline in protoplast yield, which may be attributed to physiological changes in the mycelium over time, such as possible thickening of the cell wall that could hinder enzymatic digestion [33]. Thus, younger, actively growing mycelium appears more favorable for efficient protoplast preparation.

Protoplast quality and quantity are the key factors for successful genetic transformation, especially in model and pathogenic fungi [34,35,36,37]. However, no universal protocol exists due to differences in fungal cell wall compositions [38]. In this study, protoplast preparation from *P. sclerotiorum* strain scaumcx01 was optimized using a combination of Lysing Enzymes, Yatalase, cellulase, and pectinase. This enzyme mix resulted in the highest protoplast yield of 6.72 × 10^6^ cells·mL^−1^ at 12 h, which was significantly higher compared to other treatments. Similar strategies using mixed or multiple enzyme systems have been successfully applied in other filamentous fungi. For example, in *Colletotrichum falcatum*, a combination of two enzymes has been successfully utilized to achieve protoplast formation [39]. In contrast, single enzymes treatments have been mostly employed in other filamentous fungi studies. For instance, in *Penicillium brevicompactum*, protoplast formation was efficiently carried out simply using a lytic enzyme [12], whereas the Yatalase enzyme was used in the *A. sclerotigenum* strain F-1392 [40]. In a similar study, protoplasts of *Penicillium expansum* and *Penicillium griseoroseum* both were made in the presence of pectinase [41]. While these studies focused on multiple enzymes treatments, our results demonstrate that using a combination of enzymes leads to more efficient degradation of the cell wall, significantly enhancing protoplast viability and achieving the highest recovery compared to single enzyme treatments (Figure 1e).

Mycelial age, lytic enzymes, osmotic stabilizers, temperature, and incubation time are the main factors in protoplast isolation [33,42]. Different osmotic stabilizers may be more productive depending on the organism since this depends on the composition of the cell walls in the fungal species used. Interestingly, in our study, MgSO_4_ was revealed to be the most efficacious osmotic stabilizer for *P. sclerotiorum*-scaumcx01, contrary to the common assumption that reported highly used osmotic stabilizers, such as KCl or NaCl, are more effective (Figure 1b). This aligns with findings from Chung and Park [43], who reported that MgSO_4_ yielded the highest protoplast regeneration frequency in *Penicillium verruculosum*. Conversely, in species like *Penicillium digitatum* and *Penicillium brevicompactum*, NaCl has proven equally effective for protoplast isolation, suggesting that the choice of osmotic stabilizer may vary depending on the fungal species [12,44]. The effect of enzymatic hydrolysis time and temperature was limited to osmotic stabilizers. Enzymatic digestion time was optimized and found to be less than 9–14 h in the osmotic medium for a protoplast yield. The maximum protoplast yield reached 6.72 × 10^6^ cells·mL^−1^ at 12 h (Figure 1c). Here, a transformation assay was applied to integrate the hygromycin B resistance gene into the genome of *P. sclerotiorum* strain scaumcx01. To obtain an effective genetic transformation, it is important to determine the minimal inhibitory concentrations (MICs) of the selective agents [45]. The hygromycin B selective agent was shown to be efficacious towards the regenerated protoplasts of strain scaumcx01, with the MIC of hygromycin B being 70 µg/mL as no growth of the mutants was detected at concentrations higher than the ended limit (Figure 3).

The use of PEG-mediated transformation can efficiently mediate protoplast transformation to genetic modify a fungal cell line by foreign DNA [46]. With pCT74-*GFP* plasmid-based PEG-mediated protoplast transformation, we first transferred *GFP* gene to strain scaumcx01 in this study. This process enabled DNA uptake in PEG 8000, with colonies expressing *GFP* selected and verified by PCR for gene integration and expression (Figure 4d). Our findings demonstrate the potential use of the protoplast transformation technique for *P. sclerotiorum*-scaumcx01 genetic alteration. Choosing the right plasmid is one of the most crucial aspects of fungal genetic transformation [45]. pCT74, a popular plasmid for filamentous fungal *GFP* expression, was used [46]. This approach has been effective in the past, as demonstrated by the transformation of *Sclerotinia sclerotiorum* using the same plasmid and *Trichoderma atroviride* [47,48]. Confocal microscopy was used to assess both the localization and expression of the *GFP* in *P. sclerotiorum*-scaumcx01, and provided resolution of *GFP*-tagged structures (Figure 5). Both were expressed at a high intensity, as seen in spore fluorescence at various transformants with fluorescence microscopy. Young hyphae exhibited intense *GFP* signals, while older hyphae displayed lower fluorescence (Figure 6). Interestingly, there were substantial morphological and metabolite differences between the *GFP*-tagged strain scaumcx01 and the wild-type strain, which indicates the possible effect of *GFP* tagging on the organism’s metabolism (Figure 7). Notably, several metabolites related to glycerophospholipid metabolism exhibited substantial differences, suggesting that this pathway is highly influenced by the genetic alteration (Figure 7). These observations demonstrate that fluorescent protein tagging of pCT74-*GFP* alters lipid-associated metabolism and reveals a wider effect on fungal physiology. To date, there is limited research on how *GFP*-tagging may influence the metabolome in *Penicillium* species. Further studies, such as transcriptomic and targeted metabolomic analyses, are needed to clarify whether these metabolic shifts are due to *GFP* expression itself, plasmid integration, or a stress response to the transformation procedure.

To study the interaction of fungus with a plant, we used fluorescence microscopy to determine the ability of the strain scaumcx01 labeled with *GFP* to colonize the plants. Stable expression of *GFP* in the transformants indicated successful labeling of the fungus, and none was found in the wild-type strain (Figure 8). These results highlight the need for enzymatic pretreatment prior to spread of the *GFP*-expressing scaumcx01 strain colonizing the plant roots. This finding is consistent with prior studies that have shown that the enzymatic modification of plant tissues, such as cellulase, can promote fungal attachment and invasion [49]. Microscopically, *GFP*-labeled fungal structures were primarily detected in the root epidermal layers and intercellular spaces through fluorescence microscopy, with the strongest fluorescence detected at the hair regions of roots. This indicates that the enzymatic pretreatment enabled fungal entry through their natural openings, such as the root hairs and elongation zones with robust cell wall remodeling. Other studies, including that by Panicker and Sayyed [50], have also reported that freshly harvested roots treated with enzymes (such as cellulase) promote greater fungal attachment and colonization compared to untreated roots. Our method is a useful tool for investigating fungal–plant interactions and will improve our understanding of the mechanisms underlying the colonization of plant tissue by fungi. Nonetheless, future studies should focus on thoroughly elucidating the mechanisms of these interactions and on assessing the long-term effects of fungal colonization in terms of plant health and productivity.

## 5. Conclusions

In conclusion, we successfully developed a protoplast-based transformation system for *P. sclerotiorum*-scaumcx01, which serves as an effective tool for genetic manipulation, enabling stable *GFP* expression in transgenic fungi over multiple generations. Optimization of protoplast preparation demonstrated significant effects of fungal age, enzyme combinations, and osmotic stabilizers on yields, with the highest yields achieved using a mixture of enzymes. This method induced notable metabolic shifts, particularly in glycerophospholipid metabolism. Additionally, the *GFP*-labeled strain was able to colonize tomato roots, especially after enzymatic seed treatment, suggesting surface-level interactions that may support nutrient exchange or promote plant growth. Further studies are needed to elucidate the molecular mechanisms underlying these metabolic changes, the mechanisms of plant–fungal interactions, and the potential for utilizing this transformation system to improve agricultural products.

## Figures and Tables

**Figure 1 jof-11-00386-f001:**
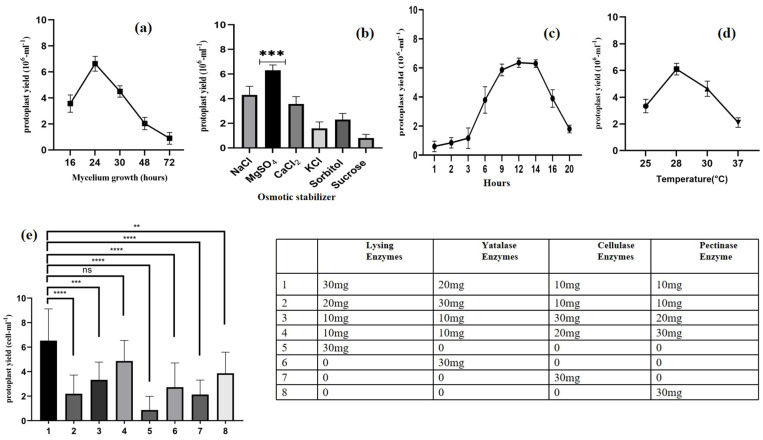
Optimization of protoplast preparation in *P. sclerotiorum*-scaumcx01: (**a**) mycelium growth, (**b**) osmotic stabilizer, (**c**) temperature, (**d**) time, and (**e**) enzyme combinations (including enzyme combination table) at significance indicated by *** *p* < 0.05; ** *p* < 0.01; **** *p* < 0.0001; ns = not significant.

**Figure 2 jof-11-00386-f002:**
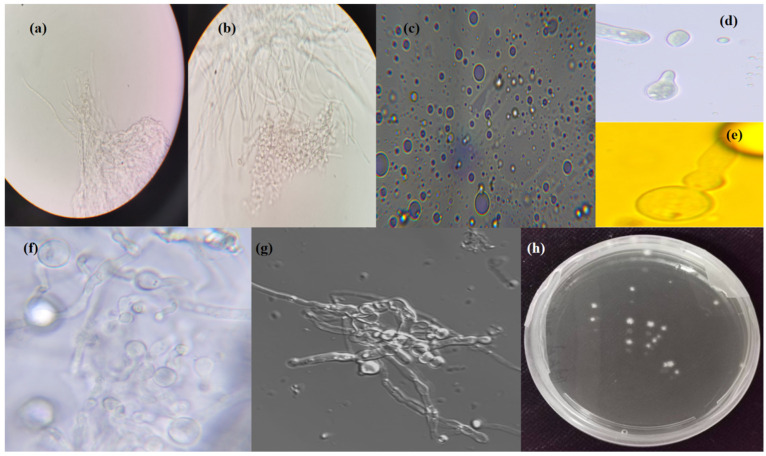
Release (**a**–**c**), purification, and regeneration (**d**–**h**) of protoplasts from *P. sclerotiorum*-scaumcx01. (**a**) Start releasing protoplast from hyphae at 12 h; (**b**) at 14 h; (**c**) purified protoplasts; (**d**) synchronous protoplast budding; (**e**) new buds continuously came out; (**f**) regeneration by a protrusion from the side to form a new protoplast; (**g**) regeneration by mycelium growing directly from the protoplast; (**h**) visible colonies.

**Figure 3 jof-11-00386-f003:**
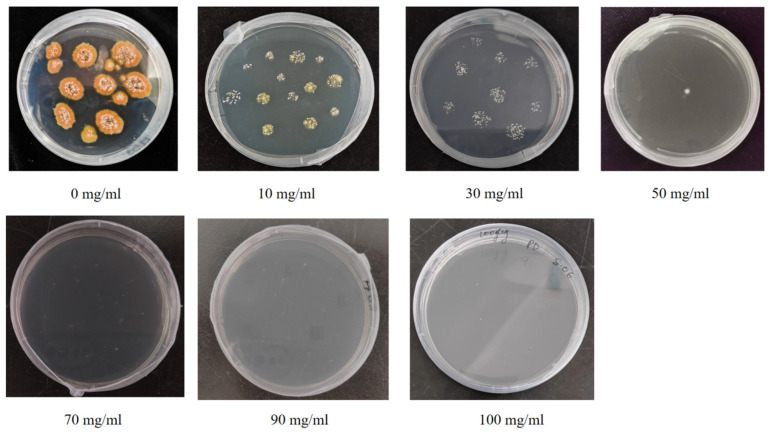
Sensitivity of *P. sclerotiorum*-scaumcx01 protoplasts to hygromycin B at various concentrations on TB3. Experiment was performed in triplicate and three replicates per treatment were conducted.

**Figure 4 jof-11-00386-f004:**
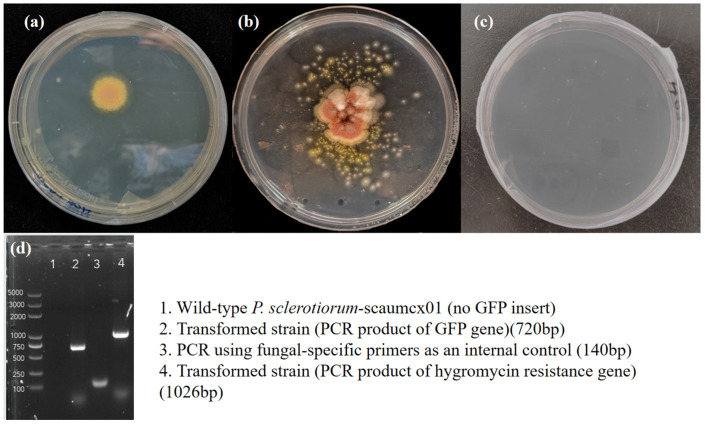
(**a**) Protoplasts transformed with pCT74-*GFP* showing growth on PDA with hygromycin B. (**b**) Wild-type strain grown on PDA without hygromycin B (growth control). (**c**) Wild-type protoplasts on PDA with hygromycin B showing no growth (negative control). (**d**) PCR confirmation of transformation.

**Figure 5 jof-11-00386-f005:**
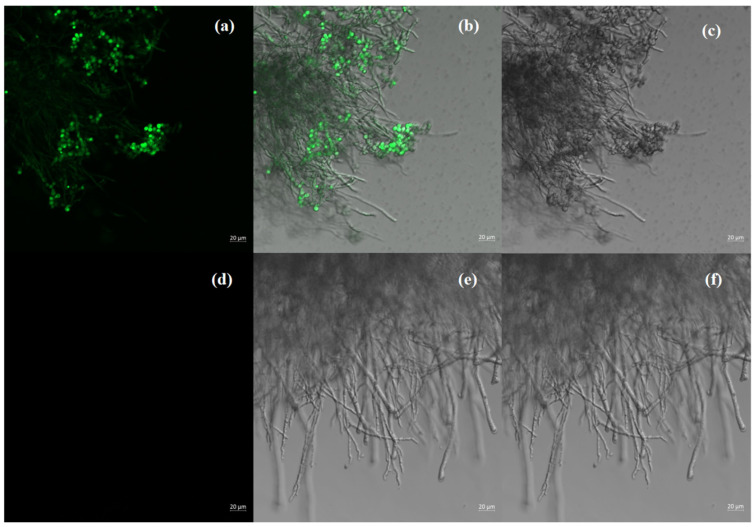
Fluorescence microscopy analysis for detection of *GFP* transformants in *P. sclerotiorum*-scaumcx01 transformed with pCT74 after five passages on PDA medium containing hygromycin (**a**,**b**) (transformants) and (**d**,**e**) (wild-type): observation of mycelium under blue light excitation. (**c**) (transformants) and (**f**) (wild-type): observation of mycelium under a bright field. Scale bars represent 20 µm.

**Figure 6 jof-11-00386-f006:**
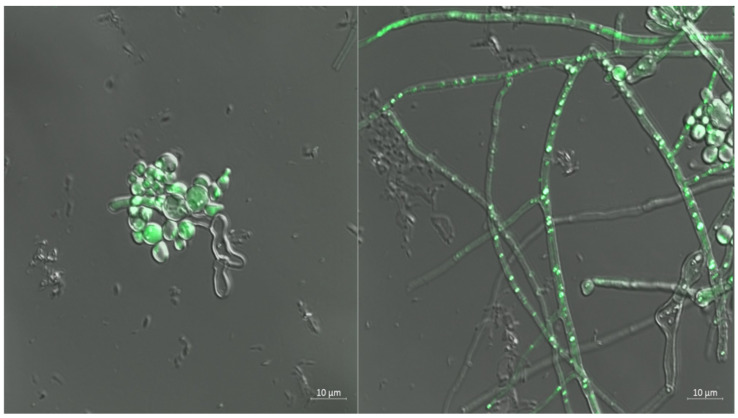
Localization of *GFP* fluorescence in spores and mycelium of *P. sclerotiorum*-scaumcx01, demonstrating its distribution in fungal structures as observed through fluorescence microscopy analysis.

**Figure 7 jof-11-00386-f007:**
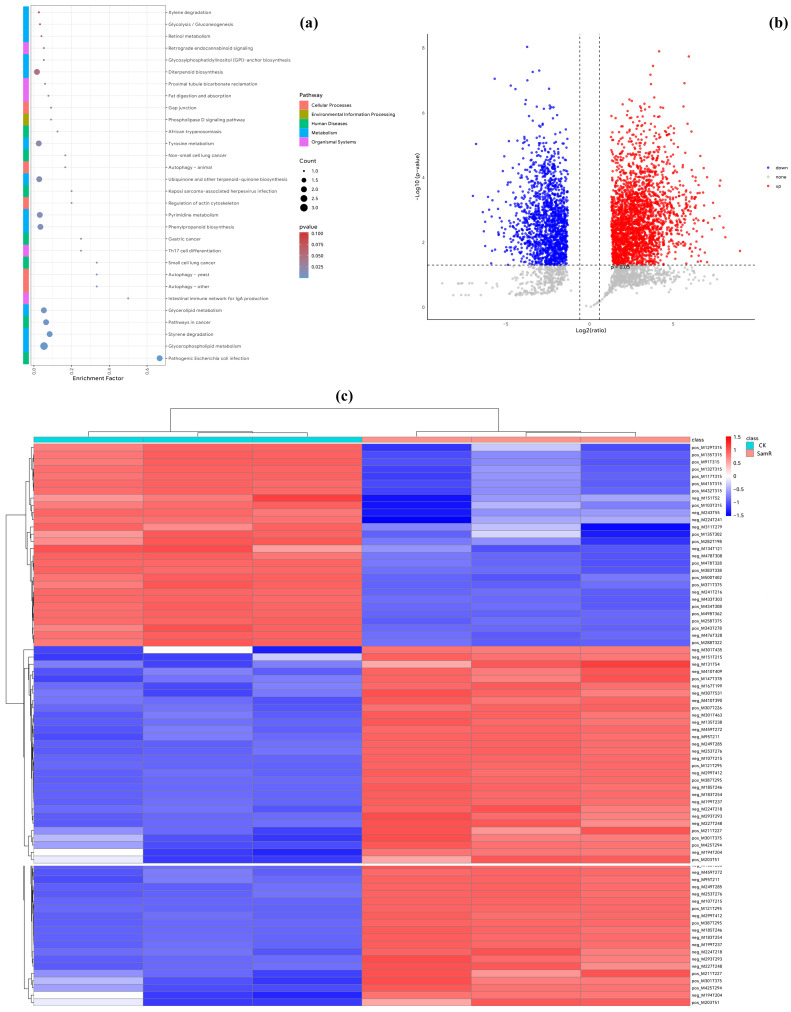
(**a**) KEGG pathway enrichment analysis revealing key metabolic pathways impacted by *GFP* tagging; (**b**) volcano plot highlighting significantly altered metabolites between wild-type and *GFP*-tagged strains of scaumcx01; (**c**) heatmap illustrating differential expression of metabolites between wild-type and *GFP*-tagged strains of scaumcx01.

**Figure 8 jof-11-00386-f008:**
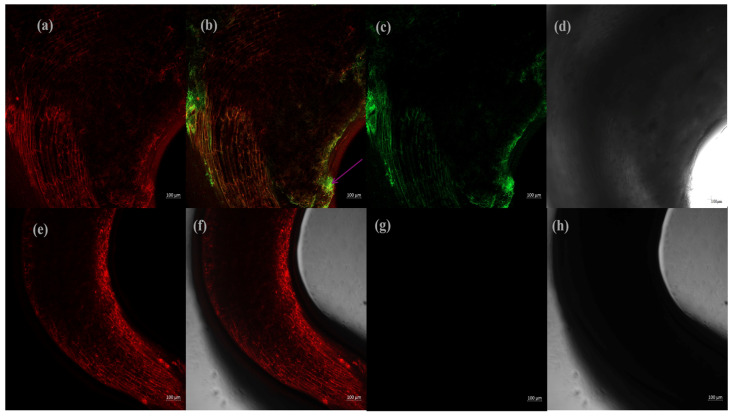
Colonization of tomato roots by pCT74-*GFP* and wild-type strain. Confocal microscopy was employed to observe colonization of tomato roots. (**a**) A section of the main root, showing root auto-fluorescence in red; (**b**) merged image of *GFP* (green) and root auto-fluorescence (red), with fungal entry points highlighted by arrowheads; (**c**) *GFP* channel (green), showing the fungus colonizing the root; (**d**) bright-field image; (**e**) root auto-fluorescence visualized in red channel (wild-type); (**f**) merged image of root auto-fluorescence and bright field (wild-type); (**g**) wild-type strain images without *GFP* expression; (**h**) bright-field image.

**Table 1 jof-11-00386-t001:** Primer sequences and amplicon sizes.

Target Gene	Sequence (5′–3′)	Amplicon Size (bp)
*GFP*	GCGACGTAAACGGCCACAAG (Forward) CCAGCAGGACCATGTGTGATCG (Reverse)	720
*hygR*	ATGAAAAAGCCTGAACTCACCGCGACGT (Forward) CGACGCCCCAGCACTCGTCCGAGGGCAA (Reverse)	1020
Fungal ITS region	GAGACGAGTTGGCAGCAGGAATAG (Forward) CCTCGCCCTTCTTGTGACTTTGG (Reverse)	140

## Data Availability

The original contributions presented in the study are included in the article/Appendix A, further inquiries can be directed to the corresponding author.

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
