# Peer review of "PEG-Mediated Protoplast Transformation of Penicillium sclerotiorum (scaumcx01): Metabolomic Shifts and Root Colonization Dynamics"

_jof, 2025, doi:10.3390/jof11050386_

Round 1
Reviewer 1 Report
In this report, Jahan and colleagues describe, for the first time, the development and optimization of protoplast transformation protocol for the fungus Penicillium sclerotiorum. In doing so, they generate a GFP-expressing strain that they is used to fungal colonization of, and influence on, tomato plant root growth. Overall, the protocols described here is well-developed and will be of interest to those studying this fungus in particular, or for those developing such protocols in their own fungus-of-interest. However, there exists several points of concern that largely reflect a lack of clarity in the writing. Other concerns are related to an overinterpretation of the data, namely with respect to the metabolomic data. Overall, while no major experiments are deemed necessary, the editorial points raised in the attached document reflect a need for substantial revision ahead of publication.
Please refer to the attached document.

Author Response
Dear Reviewer,
Thank you very much for your thoughtful and constructive comments. We have carefully revised the manuscript in response to your suggestions. All changes have been highlighted in the revised version, and detailed responses are provided in the attached response file. Your feedback helped us improve the clarity and quality of the manuscript.

Reviewer 2 Report
The manuscript "PEG-Mediated Protoplast Transformation of /Penicillium Scle- 2
rotiorum/ (scaumcx01): Metabolomic Shifts and Root Colonization Dynamics" represents a large-scale study. The authors for the first time developed a useful protoplast-based transformation system in P. sclerotiorum-scaumcx01, as a serves an effective tool for genetic manipulation with stable GFP expression in the transgenic fungus for multiple generations.
A number of complex molecular genetic studies, as well as a metabolomic analysis of wild-type and GFP-tagged scaumcx01 have been studied.
The results of the molecular genetic studies are described in detail, but the results of the metabolomic analysis are described so poorly that the results of this study are not clear and the conclusions are not obvious.
Figure 6 is very small and cannot be clearly seen.
In addition, among the minor comments, it is worth noting the poor quality of Figure 2
Author Response
Dear Reviewer,
Thank you very much for your thoughtful and constructive comments. We have carefully revised the manuscript in response to your suggestions. All changes have been highlighted in the revised version, and detailed responses are provided in the attached response file. Your feedback helped us improve the clarity and quality of the manuscript

Round 2
Reviewer 1 Report
The authors have done a nice job of addressing most of the previous concerns/suggestions. Thank you. However, a few minor comments remain. Please see the attached document for details.
Please see the attached document for details.

Author Response

(The authors gave the same response as above.)
